# Effects of a Mediterranean Diet Intervention on Maternal Stress, Well-Being, and Sleep Quality throughout Gestation—The IMPACT-BCN Trial

**DOI:** 10.3390/nu15102362

**Published:** 2023-05-18

**Authors:** Irene Casas, Ayako Nakaki, Rosalia Pascal, Sara Castro-Barquero, Lina Youssef, Mariona Genero, Leticia Benitez, Marta Larroya, Maria Laura Boutet, Giulia Casu, Alex Gomez-Gomez, Oscar J. Pozo, Ivette Morilla, Anabel Martínez-Àran, Eduard Vieta, María Dolores Gómez-Roig, Rosa Casas, Ramon Estruch, Eduard Gratacos, Fàtima Crispi, Francesca Crovetto

**Affiliations:** 1BCNatal|Fetal Medicine Research Center (Hospital Clínic and Hospital Sant Joan de Déu), University of Barcelona, 08028 Barcelona, Spain; 2Institut de Recerca August Pi Sunyer (IDIBAPS), 08036 Barcelona, Spain; 3Primary Care Interventions to Prevent Maternal and Child Chronic Diseases of Perinatal and Development Origin, RD21/0012/0001, Instituto de Salud Carlos III, 28040 Barcelona, Spain; 4Institut de Recerca Sant Joan de Deu (IRSJD), 08950 Barcelona, Spain; 5Department of Internal Medicine Hospital Clinic, IDIBAPS, University of Barcelona, 08036 Barcelona, Spain; 6Centro de Investigación Biomédica en Red de Fisiopatología de la Obesidad y Nutrición (CIBEROBN), 28029 Madrid, Spain; 7Josep Carreras Leukaemia Research Institute, Hospital Clínic, University of Barcelona Campus, 08036 Barcelona, Spain; 8Integrative Pharmacology & Systems Neuroscience Group, IMIM (Hospital del Mar Medical Research Institute), 08003 Barcelona, Spain; 9Department of Psychiatry and Psychology, Hospital Clinic, Neuroscience Institute, IDIBAPS, University of Barcelona, CIBERSAM, 08035 Barcelona, Spain; 10Centre for Biomedical Research on Rare Diseases (CIBER-ER), 28029 Madrid, Spain

**Keywords:** Mediterranean diet, pregnancy, anxiety, well-being, sleep quality

## Abstract

Stress and anxiety are frequent occurrences among pregnant women. We aimed to evaluate the effects of a Mediterranean diet intervention during pregnancy on maternal stress, well-being, and sleep quality throughout gestation. In a randomized clinical trial, 1221 high-risk pregnant women were randomly allocated into three groups at 19–23 weeks’ gestation: a Mediterranean diet intervention, a Mindfulness-Based Stress Reduction program, or usual care. All women who provided self-reported life-style questionnaires to measure their anxiety (State Trait Anxiety Inventory (STAI), Perceived Stress Scale (PSS)), well-being (WHO Five Well Being Index (WHO-5)), and sleep quality (Pittsburgh sleep quality index (PSQI)) at enrollment and at the end of the intervention (34–36 weeks) were included. In a random subgroup of 106 women, the levels of cortisol and related metabolites were also measured. At the end of the intervention (34–36 weeks), participants in the Mediterranean diet group had significantly lower perceived stress and anxiety scores (PSS mean (SE) 15.9 (0.4) vs. 17.0 (0.4), *p* = 0.035; STAI-anxiety mean (SE) 13.6 (0.4) vs. 15.8 (0.5), *p* = 0.004) and better sleep quality (PSQI mean 7.0 ± 0.2 SE vs. 7.9 ± 0.2 SE, *p* = 0.001) compared to usual care. As compared to usual care, women in the Mediterranean diet group also had a more significant increase in their 24 h urinary cortisone/cortisol ratio during gestation (mean 1.7 ± SE 0.1 vs. 1.3 ± SE 0.1, *p* < 0.001). A Mediterranean diet intervention during pregnancy is associated with a significant reduction in maternal anxiety and stress, and improvements in sleep quality throughout gestation.

## 1. Introduction

The Mediterranean diet (MedDiet) has several positive effects on individual health: randomized trials demonstrated its contribution to improved cardiovascular profiles and reduced major cardiovascular events in individuals at risk of [1] diabetes, inflammatory-based disorders, cancer, and cognitive decline [2,3,4]. Additionally, there has been increasing interest of the effects of a MedDiet on mental health, stress, and quality of life in general [5]. The role of the diet, particularly the MedDiet, in the development of mental disorders, has become a recent research focus over the past decade [6]. Several studies evaluated the effect of a MedDiet intervention on the reduction in depressive symptoms and the improvement in quality of life in individuals with major depressive disorders [7,8]. In a secondary analysis of the PREvención con DIeta MEDiterránea (PREDIMED) study, a reduced risk in depression was observed in participants with type 2 diabetes allocated to the group receiving a MedDiet supplemented by nuts (hazard ratio 0.59 (95% confidence interval (CI) 0.36 to 0.98)) [9]. A recent review based on 37 studies confirmed the association between (poly)phenols consumption and the risk of depression, and a reduction in the severity of depressive symptoms [10]. Some authors hypothesized that a high-quality diet, rich in fiber, antioxidant dietary components and omega-3-polyunsaturated fatty acids, may be linked to a reduced risk of depression, anxiety, and stress [11], which could provide new potential methods for the treatment and prevention of mental disorders in general. Moreover, it has been described that a dysregulated redox signaling is a key factor in the pathophysiology of mental disorders, especially in depression, and increased reactive oxygen and nitrogen species were observed in these patients [12,13].

Stress and anxiety are frequent occurrences among pregnant women. Peripartum anxiety disorders are more prevalent than previously thought, as 1 in 5 women can suffer from them [14]. Mental disorders can appear before pregnancy, with a changing course during pregnancy and postpartum. These findings highlight the need for screenings for stress-related disorders and education by different health professionals from the early stages of pregnancy. Several studies have shown the effectiveness of non-pharmacological treatments in the improvement in stress and other mental disorders during pregnancy, such as mindfulness meditation, biofeedback, or exercise such as yoga [15]. However, there is paucity of data regarding the dietary approach to these conditions during pregnancy. Interestingly, a recent observational study revealed an association between the MedDiet and anxiety [16]. Moreover, the production of reactive oxygen and nitrogen species production, as well as individual antioxidant capacity, is influenced by several dietary factors. A dietary intervention promoting plant-based foods that are rich in antioxidants, such fruits, vegetables, extra-virgin olive oil, and whole-grain cereals, may modulate the individual antioxidant capacity, explaining the improvements in mental wellbeing [12]. Thus, randomized clinical trials are needed to establish the potential effects of dietary patterns on mental health, avoiding the confusion attributed to the co-occurrence of other lifestyle-related and sociodemographic factors.

During pregnancy, evidence has been provided regarding the potential beneficial effects that structured dietary interventions based on a MedDiet can have, not only on pregnant women [9,13,14], but also their offspring and the pregnancy itself. In a recent randomized clinical trial, pregnant individuals at high risk for small-for-gestational-age newborns (SGA) who followed a structured MedDiet intervention significantly reduced the incidence of newborns being born small (with birth weight below the 10th percentile) and other perinatal complications [17]. However, the influence of MedDiet on maternal wellbeing during pregnancy remains to be determined.

The present study aimed to evaluate the influence of a structured intervention during pregnancy based on a MedDiet on maternal stress and anxiety, mindful state, quality of life and sleep.

## 2. Materials and Methods

### 2.1. Study Design, Population and Ethics

Improving Mothers for a better PrenAtal Care Trial BarCeloNa (IMPACT BCN) was a parallel, unblinded randomized clinical trial conducted at BCNatal (Hospital Clínic and Hospital Sant Joan de Déu), a large referral center for maternal–fetal and neonatal medicine in Barcelona, Spain. Details of the trial are provided in the protocol of the study [18], approved by the Institutional Review Board (HCB-2016-0830) before any participant enrolment. All individuals who agreed to participate provided written informed consent before randomization. Participants were screened for eligibility during routine second trimester ultrasound scans (19–23.6 weeks of gestation) for being at high risk of developing an SGA newborn [19], and were randomly assigned 1:1:1, based on a computerized random number generator, to one of the three study groups: a MedDiet supplemented with extra-virgin olive oil and walnuts; a stress reduction intervention based on the Mindfulness-Based Stress Reduction (MBSR) program; or usual care without any intervention (control group). For this specific study, only women belonging to the group of MedDiet and usual care who provided lifestyle questionnaires were included. The trial was registered in ClinicalTrials.gov Identifier: NCT03166332.

### 2.2. Interventions and Measurements

#### 2.2.1. Mediterranean Diet Program

The dietary intervention, adapted from the PREDIMED trial [20], aimed to change the general dietary pattern instead of focusing on changes in single foods or macronutrients. Participants were encouraged to increase their intake of whole-grain cereals (≥5 servings/d); vegetables and dairy products (≥3 servings/d); fresh fruit (≥2 servings/d); and legumes, nuts, fish, and white meat (≥3 servings/week), as well as increasing their olive oil use for cooking and dressings. To achieve a personalized goal, personal and individual recommendations were introduced to the participant’s diet according to height, weight, culture, and dietary preferences. Dieticians conducted 30 min face-to-face interviews at enrollment and monthly until the end of intervention (34–36 weeks’ gestation). Two weeks following each face-to-face visit, participants underwent telephone interviews. In addition, all participants received extra-virgin olive oil (2 L every month) and 15 g of walnuts per day (450 g every month) at no cost. Additional details of the intervention are provided elsewhere [18]. No intervention or advice regarding mental health, well-being, anxiety, stress, or sleep quality were provided to the participants allocated to the Mediterranean diet group.

#### 2.2.2. Usual Care (Control Group)

Women randomized into this group received usual pregnancy care as per institutional protocols (no intervention), and lifestyle questionnaires were collected at enrollment and at the end of intervention (34–36 week’s gestation). No intervention or advice regarding mental health, well-being, anxiety, stress, or sleep quality were provided to the participants allocated to the control group.

### 2.3. Outcomes

In this trial sub-analysis, the main aim was to investigate the influence of a Mediterranean diet intervention program during pregnancy on maternal stress, anxiety, well-being, mindful state, and sleep quality. Additionally, in a randomly selected subgroup of participants, the levels of cortisol, cortisone and other intermediate related metabolites were measured at the beginning and at the end of the intervention in 24 h urine samples.

### 2.4. Data Collection

The data of participants included in the study were anonymized and entered in an electronic case report form. Investigators collected maternal sociodemographic and clinical data.

All individuals included in the trial had a baseline visit (19–23 weeks of gestation) and a final visit (34–36 weeks of gestation) with a trained dietitian to assess their diet using a validated 151-item food-frequency questionnaire [21], 7-day dietary registry and the 17-item MedDiet adapted to pregnancy adherence score (score range: 0–17). All participants also provided self-report lifestyle questionnaires to measure their anxiety and stress (State-trait Anxiety Inventory (STAI) Anxiety and Personality [22], range 0–80); Perceived Stress Scale (PSS) [23], range 0–40; well-being (WHO Five Well Being Index (WHO-5) [24], range 0–100); mindful state (WHO Five Facet Mindfulness Questionnaire (FFMQ) [25], range 8–40 for the observation, description, awareness, and nonjudgmental facets, respectively, and range 7–35 for nonreactivity facet); sleep quality (Pittsburgh Sleep Quality Index (PSQI) [26], range 0–21). The questionnaires were carried out at enrollment (baseline punctuation) and at 34–36 weeks of gestation (final punctuation). Abnormal scores were considered the 75th percentile of the baseline scores of each questionnaire in the usual care group, except for the WHO-5 questionnaire, which presents a previously reported cut-off point that defines optimum mental well-being as a score greater than 52 [27].

### 2.5. Sample Collection

In a subgroup of randomly selected participants from each study group (excluding those receiving corticosteroid treatment), the 24 h urinary cortisone and cortisol metabolites were measured at the baseline and final assessment and analyzed by a validated method based on liquid chromatography-tandem mass spectrometry (LC-MS/MS) [28]. The activity of 11β-Hydroxysteroid Dehydrogenase Type 2 was estimated by the cortisone/cortisol ratio.

### 2.6. Statistical Analysis

Clinical data are presented as mean (standard deviation (SD) or standard error (SE)), median (interquartile range (IQR)) or number (percentage), as appropriate. The methods of statistical analyses used for the comparison of clinical and perinatal characteristics included Student’s *t*-test, ANOVA or ANCOVA with baseline adjustments for continuous variables and X^2^ test for categorical variables. Differences were considered significant when *p*-value < 0.05. Statistical analyses were performed using the Statistical Package for the Social Sciences statistical software package version 27 (SPSS Inc., Chicago, IL, USA).

## 3. Results

### 3.1. Study Population and Pregnancy Outcomes

Within these patients, after excluding those that did not provide lifestyle questionnaires to measure their anxiety and stress, mindful state and sleep quality, a population of 680 individuals was considered (*n* = 331 for Mediterranean diet, *n* = 349 for usual care), as reported in Figure 1.

Baseline characteristics of the study population are shown in Table 1 with no differences between study groups. Pregnancy and perinatal outcomes are shown in Appendix A, with no significant differences between groups apart from the prevalence of SGA newborns, as reported in the main outcome of the trial [17].

### 3.2. Effects of Mediterranean Diet on Stress, Anxiety, Well-Being, Sleep Quality and Mindful State

#### 3.2.1. Life-Style Questionnaires

Table 2 displays baseline and final life-style questionnaire scores on stress, anxiety, well-being, sleep quality, and mindful state between study groups, and Table 3 reports the percentage of high/poor scores at the final assessment. Perceived stress, anxiety and poor sleep quality increased throughout gestation in all study groups (Figure 2). At the end of the intervention, participants in the Mediterranean diet group showed significantly lower levels of perceived stress as compared to patients undergoing usual care, as shown in Figure 2A (mean difference −0.85 (−1.63 to −0.06), *p* = 0.035). Similarly, the Mediterranean diet group presented significantly lower final anxiety scores compared to the non-intervention group (mean 13.6 ± 0.4 SE vs. 15.8 ± 0.5, *p* < 0.004) (Figure 2B), with a lower frequency of high anxiety scores (*n* = 58, 17.9% vs. *n* = 87, 25.4%, *p* = 0.020), as reported in Table 3. Aligned with the previous findings, women’s sleep quality improved following the Mediterranean diet intervention compared to controls (PSQI mean 7.0 ± 0.2 SE vs. 7.9 ± 0.2 SE, *p* = 0.001) (see Table 2 and Figure 2C).

Regarding the well-being questionnaire, 19.8% (*n* = 65) of women from the Mediterranean diet group presented with poor well-being as compared to 27.5% (*n* = 95) in the control group (*p* = 0.02), revealing better well-being (see Table 3 and Figure 3). No significant differences between groups were observed with the mindful state questionnaire (Table 2). Changes in key foods and nutrient intake during intervention are shown in Appendix A.

#### 3.2.2. Cortisol Assessment

The baseline 24 h urinary cortisone/cortisol ratio in 106 participants was similar between groups and increased during gestation. This increase was more pronounced in the Mediterranean diet group compared to usual care (mean 1.7± SE 0.1 vs. mean 1.3 ± SE 0.1, *p* < 0.001) (Table 4). At final assessment, Mediterranean diet participants showed higher levels of total cortisone concentration (mean 134.7± SE 8.3 vs. mean 111.5 ± SE 7.7, *p* = 0.012) and percentage (mean 2.9± SE 0.1 vs. mean 2.4 ± SE 0.1, *p* = 0.002), and lower levels of the 5β-tetrahydrocortisone/Cortisone (mean 16.8 ± SE 1.2 vs. mean 21.4 ± SE 1.4, *p* = 0.032) compared to the control group.

## 4. Discussion

In this randomized clinical trial that involved pregnant women at high risk for an SGA newborn, an intervention based on MedDiet significantly reduced maternal anxiety and stress and improved well-being and sleep quality. These effects were revealed by self-reported stress questionnaires and biomarkers, as reflected by the increased estimated activity of a cortisol-deactivating enzyme.

Interest in mental health and care has grown exponentially in recent years and associations between healthy dietary patterns and mental health parameters have been reported. Jacka et al. conducted a randomized controlled trial to investigate the efficacy of a dietary intervention based on the MedDiet for the treatment of symptoms related to major depressive episodes in subjects with Major Depressive Disorder, independently of other factors such as physical activity, smoking habit, or weight loss [29]. The MedDiet group showed significantly greater improvements in symptoms of depression compared to the control group. In addition, other studies have evidenced that a lower incidence of depression incidence was significantly correlated with increasing adherence to MedDiet [7]. Additionally, in the PREDIMED study, a preventive effect for depression was found for the MedDiet in participants with type 2 diabetes [9]. Specifically, participants with type 2 diabetes allocated to the MedDiet supplemented with nuts group showed a 40% lower risk of depression compared to the control arm.

However, the evidence about the effects of dietary interventions on mental health during pregnancy is limited. Our study reveals that following the MedDiet during pregnancy is associated with a reduction in maternal anxiety/stress, together with an increase in the cortisol-deactivating enzyme. These findings are in line with previous data. In a recent study, Papandreou et al. conducted a randomized clinical trial with 40 pregnant women incorporating MedDiet recommendations into the Clinical Decision Support Systems, showing an improvement in nutritional status and reduction in health-related anxiety and depression [30]. Similarly, a longitudinal study with 152 pregnant women showed that higher adherence to the MedDiet was inversely associated with anxiety and directly associated with well-being [16]. Moreover, these associations were significant for some key foods of the MedDiet, specifically whole-grain cereals, fruits and vegetables, extra-virgin olive oil and nuts [16], food sources of dietary antioxidants whose consumption was encouraged during the intervention in our study. Aligned with our findings, other healthy dietary patterns promoting healthy foods not based on the MedDiet were associated with lower depression during pregnancy [31,32,33]. Nevertheless, in observational and cohort studies with pregnant women, some specific foods have been identified as protective against mental disorders (including depression and anxiety), including whole-grain cereals, fruits, and beans. In contrast, other foods are associated with higher risk, including ultra-processed foods such as pastries, red and processed meat, margarine, and artificial juices [16,34]. Additionally, it has been postulated that levels of depression tend to increase throughout pregnancy, highlighting the importance of structured dietary interventions to improve overall diet quality during pregnancy [33,35].

In addition to its beneficial effects on anxiety and stress, our study first demonstrates an improvement in maternal well-being and sleep quality with MedDiet. The association between higher MedDiet adherence and subjective well-being has been found in observational studies [36]. In the case of sleep quality, a longitudinal study with 150 pregnant women assessed the association between MedDiet adherence and the Pittsburgh Sleep Quality Index, showing an association between higher MedDiet adherence and better sleep quality at 16- and 34-week’s gestation, results aligned with our findings [37]. It should also be considered the burden that women go through during pregnancy may affect their mental health; research often does not recognize the multiple competing demands on women, specifically during pregnancy. However, to our knowledge, the present study is the first randomized clinical trial with a structured intervention based on a MedDiet adapted to pregnancy to evaluate well-being and sleep quality.

Several biological mechanisms have been postulated regarding the relationship between diet and mental health. First, it should be noted that the MedDiet is an easy-to-follow dietary pattern and is not only a healthy diet but also promotes a healthy lifestyle, including cultural and lifestyle elements such as conviviality, seasonality, traditional recipes, physical activity, and culinary activities [38]. These behavioral changes related to lifestyle may also have a therapeutic benefit [29]. Second, the role of diet in mental health may be mediated by inflammatory and oxidative stress pathways [12,13], the modulation of gut microbiota [39] and brain plasticity [40]. A low production of brain-derived neurotrophic factor, a peptide implicated in synaptic plasticity and neuronal survival, has been observed in patients with depression [41]. Moreover, reduced brain-derived neurotrophic factor levels were observed in pregnant women with low sleep quality, as measured by the Pittsburgh Sleep Quality Index, compared to pregnant women with good sleep quality [42]. Interestingly, in a sub-group of the PREDIMED study, significantly higher plasma levels of brain-derived neurotrophic factor were observed in participants allocated to the MedDiet supplemented with nuts group compared to the control arm, whose secretion may be also modulated by diet [43]. The fatty acid profile of the MedDiet, rich in polyunsaturated fatty acids, may also promote mental health, as low polyunsaturated fatty acid intake (mainly omega-3 fatty acids) has been associated with several mental outcomes, including depression [44,45]. Thus, several dietary components, including nutrients and bioactive compounds, are required for healthy brain function and mental health, including the synergic effect between components. Therefore, dietary interventions promoting a healthy dietary pattern rather than a single nutrient may have greater benefits for mental health [46].

Important implications regarding the mental health of the mother may be expected, including a potential benefit during the postpartum period. Maternal mental health alterations, principally anxiety, are associated with several adverse outcomes for both the mother and the offspring, including postnatal depression, pre-term birth and the poor cognitive and behavioral development of the infants [47,48,49,50]. Additionally, the estimated prevalence of anxiety disorders across the perinatal period is around 21% [51]. Our results highlight the need for anxiety and stress screenings during pregnancy, nutritional education, and referrals for evaluation and treatment if necessary. Further research is needed to characterize the impact of the MedDiet on mental health during pregnancy, including the underlying mechanisms, specifically oxidative stress, and the potential benefits for the offspring’s mental health. If confirmed, the MedDiet could become an early intervention strategy for the prevention of mental disorders [52].

The major strengths of the present study include a very well-characterized population of pregnant women who followed a structured intervention in a randomized clinical trial. Moreover, the use of different validated questionnaires with clinical applicability to assess mental stress, well-being and sleep quality provided rigor and validity to the results of the study, as well as the ability to analyze various stress-related biomarkers in a subgroup of patients with the aim of measuring stress in an empirical way. The use of validated questionnaires and biomarkers may mitigate the potential misclassification of self-reported data, along with the inherent risk of inaccuracies in the measurements.

The study has some limitations. Firstly, the trial was not designed for this purpose, although maternal stress, well-being and sleep quality were prespecified in the study protocol and assessed from the beginning of the study. Secondly, we were not able to assess long-term dietary intake, including measuring diet before pregnancy or the dietary changes from the beginning of the pregnancy. Most women were of white ethnicity and middle to high socio-economical level; hence, the results should not be extrapolated to other populations with different characteristics. These findings should be considered preliminary and require replication, including reseatch involving other study populations and an evaluation of the underlying mechanisms of action.

## 5. Conclusions

In conclusion, a MedDiet intervention significantly reduces maternal anxiety and stress, as well as improving well-being and sleep quality during gestation. Considering the increasing importance of the role of mental health during pregnancy, these findings might imply the promotion of a pregnancy-adapted MedDiet among pregnant women as a powerful public health strategy.

## Figures and Tables

**Figure 1 nutrients-15-02362-f001:**
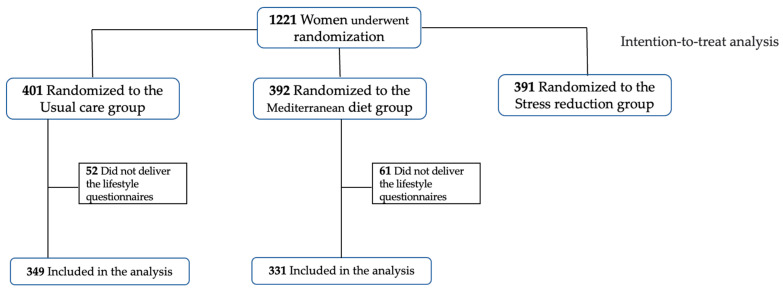
Flowchart of participants from the IMPACT BCN trial involved in the current study.

**Figure 2 nutrients-15-02362-f002:**
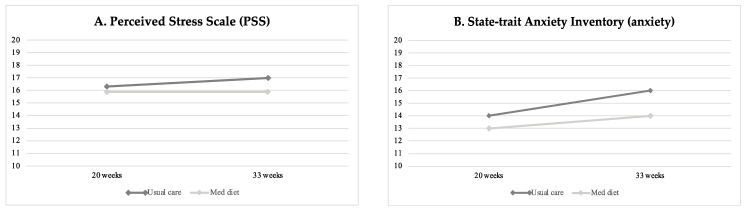
Changes in maternal stress (**A**), anxiety (**B**) and sleep quality (**C**) at baseline (20 weeks of gestation) and final (33 weeks) evaluation according to intervention groups.

**Figure 3 nutrients-15-02362-f003:**
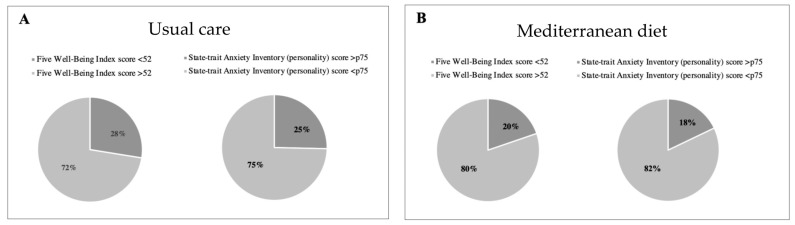
Percentage of high- vs. low-stress participants, and poor vs. good well-being (WHO-5) according to intervention groups. High stress is shown in dark grey color and defined as a State-trait Anxiety Inventory (STAI) personality score above 75th percentile in Usual care (**A**) and Mediterranean diet group (**B**). Poor well-being is shown in in dark grey color and defined as a Five Well-Being Index WHO score below 52.

**Table 1 nutrients-15-02362-t001:** Baseline characteristics of women included in the study according to intervention groups (*n* = 680).

Characteristics	Usual Care	Mediterranean Diet	*p* Value
	*n* = 349	*n* = 331	
Age at recruitment (years)	37.1 (33.3–40.5)	37.3 (34.7–40.4)	0.28
Ethnicity			
White	281 (80.5%)	269 (81.3%)	0.80
Latin	50 (14.3%)	44 (13.3%)	0.70
Afro-American	6 (1.7%)	5 (1.5%)	0.83
Asian	6 (1.7%)	7 (2.1%)	0.70
Others	6 (1.7%)	6 (1.8%)	0.93
Socio-economic status ^a^			
Low	20 (5.7%)	15 (4.5%)	0.48
Medium	106 (30.4%)	86 (26.0%)	0.20
High	223 (63.9%)	230 (69.5%)	0.12
BMI before pregnancy (Kg/m^2^)	23.7 (4.8)	24.0 (4.7)	0.60
BMI > 30 kg/m^2^ before pregnancy	39 (11.2%)	38 (11.5%)	0.90
Medical history before pregnancy			
Autoimmune disease	48 (13.8%)	39 (11.8%)	0.44
Thyroid disorders	20 (5.7%)	29 (8.8%)	0.13
Chronic hypertension	15 (4.3%)	8 (2.4%)	0.18
Diabetes Mellitus	12 (3.4%)	16 (4.8%)	0.36
Psychiatric disorders	11 (3.2%)	8 (2.4%)	0.56
Chronic kidney disease	5 (1.4%)	6 (1.8%)	0.70
Obstetric history			
Nulliparous	143 (41.0%)	145 (43.8%)	0.46
Previous placental disease	68 (19.5%)	66 (19.9%)	0.88
Previous preterm birth	9 (2.6%)	10 (3.0%)	0.73
Use of assisted reproductive technologies	92 (26.4%)	85 (25.7%)	0.84
Cigarette smoking during pregnancy	28 (8.0%)	22 (6.6%)	0.49
Alcohol intake during pregnancy	8 (2.3%)	4 (1.2%)	0.27
Drug consumption during pregnancy	1 (0.3%)	2 (0.6%)	0.77
Sports practice during pregnancy	78 (22.3%)	71 (21.5%)	0.94
Yoga or Pilates during pregnancy	73 (20.9%)	63 (19.0%)	0.54

BMI: Body mass index. Data are expressed as median (IQR) or mean (SD) or *n* (%). ^a^ socioeconomical status: low (never work or unemployed >2 years), medium (secondary studies and work), high (university studies and work).

**Table 2 nutrients-15-02362-t002:** Changes in maternal anxiety, well-being, sleep quality, and mindful state evaluated at baseline and final evaluation according to intervention groups.

		Within-Group Mean Changes		*p* §	Between-Group Changes
		Usual Care	MedDiet		MedDiet vs. Usual Care
		*n* = 349	*n* = 331		Difference
					(95% CI)
Perceived stress scale score	Baseline †	16.3 ± 7.8	15.9 ± 7.6		
	Final ‡	17.0 ± 0.4 *	15.9 ± 0.4	0.035	−0.85 (−1.63 to −0.06)
State-trait Anxiety Inventory (anxiety)	Baseline †	14.1 ± 8.8	12.9 ± 8.3		
	Final ‡	15.8 ± 0.5 **	13.6 ± 0.4 *	0.004	−1.35 (−2.28 to −0.43)
State-trait Anxiety Inventory (personality)	Baseline †	15.8. ± 9.0	14.2 ± 7.9		
	Final ‡	15.8 ± 0.5	14.0 ± 0.5	0.100	−0.68 (−1.48 to 0.13)
WHO Five Well-being index	Baseline †	62.7 ± 17.3	67.5 ± 15.2		
	Final ‡	62.9 ± 0.9	66.6 ± 0.8	0.587	0.51 (−1.32 to 2.33)
Pittsburgh Sleep Quality Index	Baseline †	6.7 ± 2.4	6.4 ± 2.1		
	Final ‡	7.9 ± 0.2 **	7.0 ± 0.2 **	0.001	−0.73 (−1.15 to −0.31)
FFMQ 1: Observation	Baseline †	23.3 ± 5.9	24.2 ± 5.6		
	Final ‡	24.0 ± 0.3	24.6 ± 0.3	0.729	0.12 (−0.57 to 0.81)
FFMQ 2: Description	Baseline †	32.1 ± 5.5	32.7 ± 4.8		
	Final ‡	31.7 ± 0.3	32.4 ± 0.3	0.273	0.35 (−0.27 to 1.37)
FFMQ 3: Awareness	Baseline †	31.3 ± 6.0	31.3 ± 6.3		
	Final ‡	30.6 ± 0.4 *	30.0 ± 0.4 **	0.280	−0.51 (−1.43 to 0.41)
FFMQ 4: Non-judgmental	Baseline †	29.9 ± 5.6	30.1 ± 5.2		
	Final ‡	30.0 ± 0.3	30.0 ± 0.3	0.994	0.00 (−0.64 to 0.64)
FFMQ 5: Non-reactivity	Baseline †	22.5 ± 4.8	22.6 ± 4.8		
	Final ‡	22.9 ± 0.2	22.5 ± 0.3	0.091	−0.55 (−1.05 to 0.08)

MedDiet: Mediterranean diet; FFMQ Five Facet. Mindfulness questionnaire. † Baseline values are observed means ± SD. ‡ Final values are baseline-adjusted (least-squares) means ± SE and comparison among groups obtained with ANCOVA analysis. * *p* < 0.05 and ** *p* < 0.001 final from baseline comparison. § ANCOVA analysis.

**Table 3 nutrients-15-02362-t003:** Frequency of women high maternal stress, poor well-being and sleep quality questionnaires score at final evaluation according to intervention groups.

Final Scores	Usual Care	Mediterranean Diet	*p* Value
	*n* = 349	*n* = 331	
Perceived Stress Scale score > p75	85 (24.4%)	80 (24.2%)	0.96
State-trait Anxiety Inventory (anxiety) score > p75 ^a^	82 (23.9%)	75 (23.1%)	0.82
State-trait Anxiety Inventory (personality) score > p75 ^a^	87 (25.4%)	58 (17.9%)	0.02
WHO Five Well-Being Index score < 52 ^b^	95 (27.5%)	65 (19.8%)	0.02
Pittsburgh Sleep Quality Index score > p75 ^c^	62 (21.8%)	44 (16.8%)	0.14

Data are expressed as *n* (%). High maternal stress/anxiety defined as Perceived Stress Scale and State-strait Anxiety Inventory scores above 75th percentile. Poor well-being defined as Five Well-Being Index score below 52. Poor sleep quality defined as Pittsburgh Sleep Quality score above 75th percentile. ^a^ Data available for 667 pregnancies. ^b^ Data available for 674 pregnancies. ^c^ Data available for 546 pregnancies.

**Table 4 nutrients-15-02362-t004:** Differences in urinary 24 h cortisol, cortisone and other related metabolites at baseline and final evaluation according to intervention group (*n* = 106).

		Within-Group Mean Changes		*p* §	Between-Group Changes
		Usual Care	MedDiet		MedDiet vs. Usual Care
		*n* = 52	*n* = 54		Difference
					(95% CI)
Total Cortisone/Total Cortisol	Baseline †	1.0 ± 0.6	1.2 ± 0.8		
	Final ‡	1.3 ± 0.1 **	1.7 ± 0.1 **	0.015	0.26 (0.05 to 0.47)
Total cortisol	Baseline †	89.9 ± 42.6	81.6 ± 36.1		
	Final ‡	89.8 ± 4.8	84.9 ± 5.3	0.619	2.66 (−7.83 to 13.16)
Total cortisol %	Baseline †	2.0 ± 0.8	2.0 ± 0.8		
	Final ‡	2.1 ± 0.1	2.0 ± 0.1	0.536	−0.08 (−0.33 to 0.17)
5β-tetrahydrocortisol	Baseline †	823.1 ± 419.3	734.4 ± 304.2		
	Final ‡	777.8 ± 54.6	766.3 ± 55.3	0.279	64.9 (−52.60 to 182.42)
5β-THF/Cortisol	Baseline †	10.0 ± 5.2	10.9 ± 5.0		
	Final ‡	9.1 ± 0.6	9.6 ± 0.7	0.774	0.19 (−1.13 to 1.52)
Total cortisone	Baseline †	85.6 ± 52.5	87.0 ± 50.1		
	Final ‡	111.5 ± 7.7 *	134.7 ± 8.3 **	0.012	24.3 (5.45 to 43.3)
Total cortisone %	Baseline †	1.9 ± 1.0	1.9 ± 0.7		
	Final ‡	2.4 ± 0.1 **	2.9 ± 0.1 **	0.002	0.47 (0.18 to 0.78)
5β-tetrahydrocortisone %	Baseline †	2185.2 ± 1189.3	1961.1 ± 973.2		
	Final ‡	2209.3 ± 171.2	2196.5 ± 184.4	0.627	111.0 (−336.96 to 558.99)
5β-THE/Cortisone	Baseline †	29.8 ± 15.5	26.3 ± 14.8		
	Final ‡	21.4 ± 1.4 **	16.8 ± 1.2 **	0.032	−3.39 (−6.49 to −0.30)

5β-THF/Cortisol: 5β-tetrahydrocortisol/Cortisol; 5β-THE/Cortisone: 5β-tetrahydrocortisone/Cortisone. † Baseline values are observed means ± SD. ‡ Final values are baseline-adjusted (least-squares) means ± SE and comparison among groups obtained with ANCOVA analysis. * *p* < 0.05 and ** *p* < 0.001 final from baseline comparison. § ANCOVA analysis.

## Data Availability

The datasets used and/or analyses during the current study are available from the corresponding author on reasonable request.

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
