# Peer review of "Effects of a Mediterranean Diet Intervention on Maternal Stress, Well-Being, and Sleep Quality throughout Gestation—The IMPACT-BCN Trial"

_nutrients, 2023, doi:10.3390/nu15102362_

Round 1

Reviewer 1 Report

The manuscript is about innovative and relevant themes, however it has some problems. In the methodology there is not enough detail about the method to treat anxiety and stress in pregnant women. Did participants in the control group receive any treatment for anxiety and stress? These questions are important to the reader and I suggest they be included.

Author Response

  • The manuscript is about innovative and relevant themes, however it has some problems. In the methodology there is not enough detail about the method to treat anxiety and stress in pregnant women. Did participants in the control group receive any treatment for anxiety and stress? These questions are important to the reader and I suggest they be included.

Answer: Thanks for the comment. This study is a sub-analysis of the main trial IMPACT BCN (Crovetto et al., JAMA. 2021;326(21):2150-2160. doi:10.1001/jama.2021.20178). Briefly, the IMPACT BCN was a randomized clinical trial were pregnant women at high-risk for small for gestational age newborns were randomly allocated to three intervention arms: an intervention based on Mediterranean diet adapted to pregnancy; an intervention based on a mindfulness-based stress reduction program or usual care group (control group without intervention). In the main results of the IMPACT BCN study published elsewhere (Crovetto et al. JAMA 2021), we reported a significant reduction in the all aspects related to anxiety and stress during pregnancy in the participants allocated in the stress reduction group, revealed by self-reported questionnaires as well as cortisol levels.

For the present study, only the Mediterranean diet and control group were included, in order to evaluate whether a structured lifestyle intervention based on Mediterranean diet may have an impact on maternal wellbeing, anxiety and/or stress compared to control group. So, in these groups of women there was not a specific treatment or intervention for anxiety/stress and we wanted to see if the diet itself could have an indirect effect on this. However, we agree with the reviewer that was not clear enough, so in order to clarify it, we added to the manuscript these sentences “No intervention or advice regarding mental health, well-being, anxiety, stress or sleep quality were provided to the participants allocated in the Mediterranean diet group”, in line 130 and “No intervention or advice regarding mental health, well-being, anxiety, stress or sleep quality were provided to the participants allocated in the control group”, line 136.

Reviewer 2 Report

Thank you for the opportunity to review this paper regarding the use of the Med diet in the prenatal period. It is an interesting concept and I have provided feedback below on how the manuscript can be improved. 

Overall feedback:

I think this is a novel study. However, it is difficult for me to review the intro and discussion as I think there needs to be clarification of the methods and results first.

Further, there needs to be an understanding that the link between mental health and diet is considered in the context of long term dietary change. I would like to see more justification that the length of this study (which is not clear) would provide enough time for it to make a difference to cognitions and neural pathways which in some cases (such as the psychiatric condition participants) could be chronic.

There also needs to be consideration of the burden that women go through during pregnancy. Research often does not recognise the multiple competing demands on women. This is an issue of equality in medical research broadly. For this reason, ensure your language is strengths based.  

Methods:

‘for being at high risk of developing an SGA newborn’– can you give examples for why these women were high risk for SGA infants. Technically this is looking at a hypothetical outcome and not describing the population. For example, were the women undernourished, have gestational hypertension, preeclampsia etc.
What was the randomisation method?

Dieticians conducted face-to-face interviews at enrolment, and monthly until the end of intervention (34-36 weeks’ gestation)’ – regarding what? What was covered in this consult, how long was the consult? Was follow up appointments or information provided?

‘Two weeks following each face-to-face visit, participants underwent telephone interviews’ – With dietitians? Again, what was covered in these consults and how long were they?

‘well-being, mindful state’ – please define these outcome variables. The current terminology is too vague

Abnormal scores were considered the 75th percentile of the baseline scores of each questionnaire in the usual care group was calculated, - please identify why this categorisation was used when the questionnaires used have validated categorisation?

Was there any measures to ensure that the med diet group complied to the diet? What percentage of compliance/macronutrient targets were set?

Results

‘ Within these patients, after excluding those that did not provide lifestyle questionnaires’ – This is confusing. Are the lifestyle questionnaires referred to here the outcome measures or screening questionnaires?

In figure 1 – did the stress reduction group not have to complete the lifestyle questionnaires?

Epidemiological characteristics – Do you mean demographics?

Where is the stress reduction group in table 1?

I am assuming that the conditions listed under ‘medical history’ were pre-existing prior to pregnancy?

A BMI>30 is not a pathological disorder.

What level of alcohol intake and smoking was considered?

What were considered autoimmune diseases and psychiatric disorders for inclusion in Table 1? I am interested in this as these are going to significantly influence the endocrine stress markers.

Along the same lines, what was considered ‘sports practice’?

It is not clear whether the baseline values had t-tests conducted for table 2

The title of the first data column in table 2 is ‘Within-group mean changes’. Are you presenting the delta values or the mean values here? This is not clear.

Table 3 - Total Cortisone/Total Cortisol – which one is it? Or is it the ratio (this is not clear for other outcomes in this table). 

This manuscript has multiple grammatical and spelling errors. I recommend the authors try proof reading the document again or seek support for English language and expression. 

Author Response

Reviewer 2

Thank you for the opportunity to review this paper regarding the use of the Med diet in the prenatal period. It is an interesting concept and I have provided feedback below on how the manuscript can be improved.

Overall feedback:

  • I think this is a novel study. However, it is difficult for me to review the intro and discussion as I think there needs to be clarification of the methods and results first.

  • Further, there needs to be an understanding that the link between mental health and diet is considered in the context of long term dietary change. I would like to see more justification that the length of this study (which is not clear) would provide enough time for it to make a difference to cognitions and neural pathways which in some cases (such as the psychiatric condition participants) could be chronic.

Answer: Thank you for your appreciation. This randomized clinical trial was design to conduct the intervention during pregnancy. Specifically, participants were included at 19-23 weeks’ gestation and the intervention finished at 34-36 weeks’ gestation. This information is included in the Abstract (line 34), and the methods section for both groups (Mediterranean diet group line 126 and control group line 136). We agree that the evaluation of long-term dietary intake or following a healthy dietary pattern may influence cognition. We included this as a limitation in line 352: “Secondly, we were not able to assess long-term dietary intake, including to measure diet before pregnancy or the dietary changes from the beginning of the pregnancy.”. However, the aim of this specific study was to evaluate if a structured dietary intervention based on Mediterranean diet during pregnancy may influence mental health.

  • There also needs to be consideration of the burden that women go through during pregnancy. Research often does not recognise the multiple competing demands on women. This is an issue of equality in medical research broadly. For this reason, ensure your language is strengths based.

Answer: We agree with the reviewer that pregnancy is a period of women’ life full of expectative but also stress for the uncertainty and high pressure from the society. We add a paragraph in the discussion (line 303).  “It should also be considered the burden that women go through during pregnancy that may affect mental health, while research often does not recognize the multiple competing demands on women, specifically during pregnancy”. However, we are sorry, but we do not understand the second part of the comment, regarding the language of the manuscript and the issue of equality in medical research (that of course we completely agree, but we do not understand which is the relation with this current study).

Methods:

  • ‘for being at high risk of developing an SGA newborn’– can you give examples for why these women were high risk for SGA infants. Technically this is looking at a hypothetical outcome and not describing the population. For example, were the women undernourished, have gestational hypertension, preeclampsia etc. What was the randomisation method?

Answer: The protocol of the trial was described elsewhere (Crovetto, et al. Trials 2021) and the reference is included in line 109. The inclusion criteria for high risk for SGA newborns were defined according to the Royal College of Obstetricians and Gyneacologists. Regarding the randomization method, participants were randomly assigned to the intervention groups with a 1:1:1 allocation as per computerized random number generator, with equal proportion in each group. This information is included in the line 109 in the methods section “Study design, Population and Ethics”, however, we added more information regarding the randomization method in line 109.

Concerning the description of the population, we included in the Table 1 information regarding pre-pregnancy condition (pre-pregnancy BMI and medical conditions), as well as other life-style factors. Moreover, the prevalence of complications during pregnancy, delivery, and neonatal outcome are described in Supplementary Material Table 1.

Aligned with previous comments, the IMPACT BCN trial was not designed for the purpose of the present study, although maternal stress, well-being and sleep quality were prespecified in the study protocol and assessed from the beginning of the study. We added this as the first limitation of our study (line 350).

  • Dieticians conducted face-to-face interviews at enrolment, and monthly until the end of intervention (34-36 weeks’ gestation)’ – regarding what? What was covered in this consult, how long was the consult? Was follow up appointments or information provided? ‘Two weeks following each face-to-face visit, participants underwent telephone interviews’ – With dietitians? Again, what was covered in these consults and how long were they?

Answer: The intervention was detailed in Methods section “Interventions and measurements”.

Trained dietitians were responsible of the dietary intervention as well as nutritional data collection. The intervention was intensive, with four face-to-face interviews and telephone interviews between face-to-face visits, to ensure the adherence to the intervention. More details about what was covered during the visits and the nutritional objectives of the intervention are described in methods section. Regarding the length of the visits, face-to-face interviews were about 30 minutes, while telephone interviews were around 10 minutes. We added in line 125 the length of the face-to-face interviews.

  • ‘well-being, mindful state’ – please define these outcome variables. The current terminology is too vague

Answer: Well-being is a questionnaire of the WHO, whereas mindful state refers to mindfulness related questionnaire (WHO Five Facet Mindfulness Questionnaire-FFMQ).

  • Abnormal scores were considered the 75th percentile of the baseline scores of each questionnaire in the usual care group was calculated, - please identify why this categorisation was used when the questionnaires used have validated categorisation?

Answer: In case of the WHO-5 questionnaire, we used the previously reported cut-off point: optimum mental well-being as a score greater than 52 [Bonnín, et al. J Affect Disord 2018]. For the remaining questionnaires, no cut-off validated for pregnant women are available.

  • Was there any measures to ensure that the med diet group complied to the diet? What percentage of compliance/macronutrient targets were set?

Answer: The main objective of the Mediterranean diet intervention arm was to increase the adherence to the 17-item Mediterranean diet score. Moreover, the changes in dietary intake including both nutrients and key-foods are detailed in Supplementary material Table 2 & 3. Significant differences were observed in most of the key-foods and nutrients, and specifically, in Mediterranean diet adherence score (mean difference Mediterranean diet group vs control: 4.26 (3.92 to 4.60) p<0.001).

Results

  • ‘ Within these patients, after excluding those that did not provide lifestyle questionnaires’ – This is confusing. Are the lifestyle questionnaires referred to here the outcome measures or screening questionnaires?

Answer: Yes, it is referring to the lifestyle questionnaire related to the outcomes, as it is detailed in the methods section “Data collection” line 145. To clarify this concept, we added in line 182 “lifestyle questionnaires to measure their anxiety and stress, mindful state and sleep quality”.

  • In figure 1 – did the stress reduction group not have to complete the lifestyle questionnaires?

Answer: Yes, but as we previously mentioned, the present study only considered the Mediterranean diet and usual care groups. Therefore, the participants allocated in the Stress reduction group were excluded for the present study. We change Figure 1 to clarify our study design.

  • Epidemiological characteristics – Do you mean demographics?

Answer: Yes, thanks for the suggestion, we changed “epidemiological characteristics” to “baseline characteristics” (Line 192).

  • Where is the stress reduction group in table 1?

Answer: As we previously mentioned, the present study only considered the Mediterranean diet and usual care groups. Therefore, the participants allocated in the Stress reduction group were excluded for the present study.

  • I am assuming that the conditions listed under ‘medical history’ were pre-existing prior to pregnancy?

Answer: Yes, we modified the tittle to “medical history before pregnancy”.

  • A BMI>30 is not a pathological disorder.

Answer: The reviewer is right, even if we indirectly referred here to obesity, that is a chronical condition. However, to make it clearer, we removed obesity from “medical condition”, and we added this data below mean BMI before pregnancy values in Table 1.

  • What level of alcohol intake and smoking was considered?

Answer: It was collected using a self-reported questionnaire about both alcohol intake and smoking during pregnancy. In Table 1 we showed the number of participants that reported “yes” to alcohol intake or smoking during pregnancy (possible answers were yes/no/stop at the beginning of the pregnancy).

However, regarding alcohol intake, in the face-to-face interviews, trained dietitians collected nutritional data, including alcohol intake. The alcohol intake was almost null (mean alcohol intake 0.18 g/day). We did not include this specific information, as no significant differences were observed between groups and participants were randomly allocated to the intervention arms.

  • What were considered autoimmune diseases and psychiatric disorders for inclusion in Table 1? I am interested in this as these are going to significantly influence the endocrine stress markers.

Answer: The presence of a major psychiatric disorders that impose doubts regarding the true patient’s willingness to participate in the study was an exclusion criteria of the trial (Crovetto et al. Trials 2021). However, previous history of mild psychiatric disorders was reported, mainly anxiety, depression, or eating disorders. Regarding autoimmune diseases, most of the autoimmune disorders were related to thyroid gland, inflammatory bowel diseases (including ulcerative colitis, Crohn’s, and celiac diseases), lupus, psoriasis, and rheumatoid arthritis, respectively. However, we did not include this specific information, due to no significant differences were observed between groups and participants were randomly allocated to the intervention arms.

  • Along the same lines, what was considered ‘sports practice’?

Answer: As previously reported, this information was collected using a self-reported questionnaire about sport practice, including yoga or Pilates, during pregnancy. In Table 1 we showed the number of participants that reported “yes” to sport practice during pregnancy (possible answers were yes/no/stop at the beginning of the pregnancy).

  • It is not clear whether the baseline values had t-tests conducted for table 2

Answer: No significant differences were observed at baseline between groups, and we showed in Table 2 the mean differences after the intervention between groups, as well as the changes in the scores pre- and post-intervention. 

  • The title of the first data column in table 2 is ‘Within-group mean changes’. Are you presenting the delta values or the mean values here? This is not clear.

Answer: As it is detailed in the table footnote, Baseline values are observed means ± SD. ‡ Final values are baseline-adjusted (least-squares) means ± SEin Line 220.

  • Table 3 - Total Cortisone/Total Cortisol – which one is it? Or is it the ratio (this is not clear for other outcomes in this table).

Answer: Excuse us, according to your question, we guess that you are referring to Table 4. In table 4, we showed the results of urinary 24-hours cortisol, cortisone, and other related metabolites. We believe that the nomenclature is clear, but please let us know if any clarification should be added.
